# Dynamic Changes and Temporal Association with Ambient Temperatures: Nonlinear Analyses of Stroke Events from a National Health Insurance Database

**DOI:** 10.3390/jcm10215041

**Published:** 2021-10-28

**Authors:** Che-Wei Lin, Po-Wei Chen, Wei-Min Liu, Jin-Yi Hsu, Yu-Lun Huang, Yu Cheng, An-Bang Liu

**Affiliations:** 1Department of Biomedical Engineering, College of Engineering, National Cheng Kung University, Tainan 701401, Taiwan; lincw@mail.ncku.edu.tw; 2Medical Device Innovation Center, National Cheng Kung University, Tainan 704302, Taiwan; 3Institute of Gerontology, College of Medicine, National Cheng Kung University, Tainan 701401, Taiwan; 4Medical Department, Hualien Tzu Chi Hospital, Buddhist Tzu Chi Medical Foundation, Hualien 970473, Taiwan; drpwchen@gmail.com; 5Department of Computer Science and Information Engineering, National Chung Cheng University, Chiayi 621301, Taiwan; wmliu@cs.ccu.edu.tw; 6Center for Aging and Health, Hualien Tzu Chi Hospital, Buddhist Tzu Chi Medical Foundation, Hualien 970473, Taiwan; elliot771013@hotmail.com; 7Department of Medicine, School of Medicine, Tzu Chi University, Hualien 970374, Taiwan; joe2468joe2468@gmail.com; 8Department of Medical Education, Taipei Tzu Chi Hospital, Buddhist Tzu Chi Medical Foundation, Taipei 231405, Taiwan; kentcheng1004@gmail.com; 9Department of Neurology, Hualien Tzu Chi Hospital, Buddhist Tzu Chi Medical Foundation and Tzu Chi University, Hualien 970473, Taiwan

**Keywords:** ambient temperature, stroke event, nonlinear analysis, temporal association

## Abstract

Background: The associations between ambient temperatures and stroke are still uncertain, although they have been widely studied. Furthermore, the impact of latitudes or climate zones on these associations is still controversial. The Tropic of Cancer passes through the middle of Taiwan and divides it into subtropical and tropical areas. Therefore, the Taiwan National Health Insurance Database can be used to study the influence of latitudes on the association between ambient temperature and stroke events. Methods: In this study, we retrieved daily stroke events from 2010 to 2015 in the New Taipei and Taipei Cities (the subtropical areas) and Kaohsiung City (the tropical area) from the National Health Insurance Research Database. Overall, 70,338 and 125,163 stroke events, including ischemic stroke and intracerebral hemorrhage, in Kaohsiung City and the Taipei Area were retrieved from the database, respectively. We also collected daily mean temperatures from the Taipei and Kaohsiung weather stations during the same period. The data were decomposed by ensemble empirical mode decomposition (EEMD) into several intrinsic mode functions (IMFs). There were consistent 6-period IMFs with intervals around 360 days in most decomposed data. Spearman’s rank correlation test showed moderate-to-strong correlations between the relevant IMFs of daily temperatures and events of stroke in both areas, which were higher in the northern area compared with those in the southern area. Conclusions: EEMD is a useful tool to demonstrate the regularity of stroke events and their associations with dynamic changes of the ambient temperature. Our results clearly demonstrate the temporal association between the ambient temperature and daily events of ischemic stroke and intracranial hemorrhage. It will contribute to planning a healthcare system for stroke seasonally. Further well-designed prospective studies are needed to elucidate the meaning of these associations.

## 1. Introduction

Cerebrovascular disease, including ischemic stroke (IS), intracerebral hemorrhage (ICH), subarachnoid hemorrhage, and miscellaneous entities is one of the most common causes of death around the world [1,2,3]. In addition to its high mortality rate, severe morbidity causes suffering for patients and places burdens on families and society [4]. The risk factors of stroke including atrial fibrillation, dyslipidemia, hypertension, and diabetes mellitus have been well-studied [5,6,7]. The association between climate and stroke incidence has been observed clinically for a long time. Previous studies have suggested that changes in ambient temperature or other meteorological parameters including air pressure, humidity, sunshine duration, and rainfall might trigger a stroke [8,9,10]. There are several hypotheses regarding the possible mechanisms of stroke related to weather changes, including increased sympathetic tone resulting in elevated blood pressure in cold weather or vigorous temperature changes [11] and dehydration by excessive perspiration causing hyperviscosity in hot weather [12,13]. However, the association between ambient temperature and stroke incidence remains inconsistent. One study reported the highest incidence in the summer [14]. In contrast, other studies reported peak incidences in the winter or spring [10,15]. Differences in studies may be due to different risk factors and health care systems in the study areas [16]. National health insurance covers most residents in Taiwan and offers a sufficient and equal health care system countrywide compared with some other countries. The claims database of the national health insurance, National Health Insurance Research Database (NHIRD), has been used for studies of risk factors, incidences, and changes in therapeutic strategies for disease by big data analysis [17]. The Tropic of Cancer passes through the center of Taiwan and divides the island into subtropical (northern) and tropical (southern) areas (Figure 1). Thus, it is a suitable model to investigate the association between daily temperature and stroke incidence and the impact of the climate or latitude on a potential association by analyzing the NHIRD in two areas with different climate zones [18,19].

Stroke is a vascular incident that mainly results from progressive atherosclerosis or vascular anomalies. Therefore, more than one stroke can occur over a lifetime. However, most studies have focused on the incidence of stroke or first-ever stroke. A prospective study showed that stroke recurrent rates were 5.7% and 22.5% within one and five years, respectively. The recurrent rates of IS were higher than those for ICH in China [20]. A hospital-based study of NHIRD found that the one-year recurrence of ischemic stroke was 9.6% and 7.8% in 2000 and 2001, respectively [21]. Therefore, when investigating these dynamic changes and temporal association with ambient temperatures, stroke events are more relevant than the incidence or case number of the first-ever stroke. The use of the first five columns of discharge diagnosis of claimed data for hospitalized patients is a valid reliable method to estimate stroke incidence in NHIRD [22]. To decrease the possibility of overestimation, we added brain image studies to the inclusion criteria to confirm the events of stroke in this study.

A previous study reported that more than 60% of Taiwanese stroke patients arrived at hospital within 24 h in 1997 [23]. Another hospital-based study conducted in Kaohsiung from June 2004 to October 2005 showed that 93% of 129 patients with acute ischemic stroke presented to emergency services within 3 h [24]. The rate has increased following promotions by the government, as well as patient and academic societies. The awareness of early stroke features, as well as convenient, fast transportation and efficient healthcare and claim systems make it feasible to obtain reliable daily events of stroke from the NHIRD.

Empirical mode decomposition (EMD) is a proposed algorithm for decomposing or analyzing nonlinear and nonstationary signals. The EMD decomposes non-stationary signals into a number of intrinsic mode functions (IMFs), each of which is a mono-component function by repetitively averaging an envelope of regional maxima and minima. Briefly, local maxima and minima are identified to generate upper and lower envelopes using a cubic spline method. Then, a mean envelope is developed by calculating the mean of the upper and lower envelopes as the first IMF (IMF1). The IMF1 is subtracted from the original signal to obtain a new series. The new signal series is processed with the same procedures and then comes to be another mean envelope as the second IMF (IMF2). The procedures are repeated until the residual becomes monotonic. Therefore, oscillating intermixed signals can be decomposed into several IMFs. Meanwhile, the domain frequency of IMF decreases during the procedures until a residual trend arises [25]. However, certain problems such as mode mixing impede the use of this algorithm. Ensemble empirical mode decomposition (EEMD) has been developed to overcome these difficult issues. After adding appropriate white noise, the original nonlinear data can be decomposed into different intrinsic mode functions (IMFs) by EMD [26]. We have used EMD and EEMD to decompose and analyze nonlinear physiological signals, including digital volume pulse, ECG, R-R intervals, and surface electromyography, to extract stable and purified unique physical signals for analysis [26,27]. In this study, we used EEMD to decompose nonstationary daily stroke events and ambient temperatures in the Taipei Area located in a subtropical zone and Kaohsiung City in a tropical zone to verify the association between stroke occurrence and temperature and the impact of climate zones on these associations.

## 2. Methods

### 2.1. Data Source and Definition of Stroke Onset

We retrieved data of stroke patients with IS or ICH aged 20 years or older who were diagnosed with stroke from the NHRID using the following criteria: 1. admitted via emergency services with a diagnostic code of stroke in any column of their discharge diagnoses (up to five); 2. having received head CT or brain MRI by emergency services or during hospitalization from January 2010 to December 2015. Stroke diagnoses were coded using the International Classification of Diseases, ninth revision with clinical modification (ICD-9-CM code) for 2010–2014 and ICD-10 in 2015, following the regulations of the insurance claims of the national health insurance. The diagnoses of IS were coded as 433.xx, 434.xx, or 436.xx, excluding 433.x0 and 434.x0. These were coded as I63, I67.8, I67.9, G46, or H34 in 2015. The diagnoses of ICH were coded as 430, 431, 432, or 852 with ICD-9-CM and I60 or I61 with ICD-10 [22,28]. The onset of stroke was the date of visiting the emergency service. The retrieved stoke patients were divided into a middle-aged (20 years to 59 years) group and an elderly group (aged equal to or older than 60 years). We obtained the daily mean temperature data for the Taipei and Kaohsiung weather stations from the Central Weather Bureau, Taiwan for January 2010–December 2015. The Institutional Review Board of Hualien Tzu Chi General Hospital approved this research protocol (IRB 107-188-C).

### 2.2. Meteorological and Geographic Background of the Study Design

Taipei, New Taipei, and Kaohsiung are big cities in Taiwan. The first two cities are located in the subtropical region, and the latter is in the tropical region. Taipei and New Taipei are close together, and more than six million people live and work there. It is not easy to identify patients’ residence using a hospital-based claim data in a disease incidence study. Therefore, we used the Taipei Area, including Taipei City and New Taipei City, to estimate the occurrence of stroke in the subtropical area in this study. Kaohsiung City is the biggest city in this region, with a population of 2.7 million. We used this city to study stroke onset in a tropical area (Figure 1). Although the distance from Kaohsiung City to the Taipei Area is only 350 km, the climates are quite different between these areas. Kaohsiung City is located in an Am (tropical, monsoon) zone, and the Taipei Area is a Cfa (temperate, no dry season, hot summer) zone, according to Köppen–Geiger climate classification [29].

### 2.3. Ensemble Empirical Mode Decomposition of Daily Stroke Incidence and Ambient Temperature

We used an EEMD algorithm to decompose the daily mean temperatures and following stroke events, as previously described. Appropriate white noise was added to the original data before decomposition. Then, the new series was decomposed into several IMFs with EMD. The decompositions were repeated with a different white noise at each timepoint. Finally, the ensemble means of the corresponding IMFs of the decomposed data were defined as the final result [30].

Empirical mode decomposition (EMD) is a versatile adaptive time-frequency data analysis method for extracting signals from data generated in noisy nonlinear and non-stationary processes in a variety of situations. However, the frequent appearance of mode mixing, which is the consequence of signal intermittency (defined as a single intrinsic mode function (IMF), either consisting of signals of widely disparate scales or a signal of a similar scale residing in different IMF components), has been reported to be one of the major drawbacks of the original EMD. The signal intermittency can not only cause ambiguity in the time-frequency distribution, but also obscure the physical meaning of individual IMFs.

To encounter this problem, ensemble empirical mode decomposition (EEMD) [31] was adopted in the current study. The EEMD method consists of an ensemble of data decompositions with added white noise and then treats the resultant mean as the final true result. The principle of EEMD is to add white noise, which populates the whole time-frequency space uniformly with the constituent components of different scales separated by a filter bank. The EEMD process is explained as follows:(1)Add a white noise series to the targeted data;(2)Decompose the data with added white noise into IMFs;(3)Repeat step 1 and step 2 again and again, but with different white noise series each time;(4)Obtain the (ensemble) means of corresponding IMFs of the decompositions as the final result.

The result of EEMD is obtained when the number in the ensemble approaches infinity (Equation (1)):(1)$$c_{i}\left(t\right)=\underset{N\to \infty }{\lim}\frac{1}{N}\sum_{k=1}^{n}\left\{c_{i,k}\left(t\right)+\alpha r_{k}\left(t\right)\right\},$$
in which $c_{i,k}\left(t\right)+\alpha r_{k}\left(t\right)$ is the *k*-th realization of the *k*-th IMF in the noise-added signal, α is the standard deviation of the added noise, and *r*(*t*) is the residual after extracting the first *k* IMF components. The number of the trials in the ensemble, *N*, has to be large.

### 2.4. Statistical Analysis

Data are expressed as the mean ± standard deviation (SD). The significance of differences in daily mean temperatures and events of stroke between Kaohsiung City and the Taipei Area were determined by the nonparametric Mann–Whitney U-test at * *P* < 0.05. Significant correlations between the daily mean temperatures and stroke events were evaluated by Spearman’s rank correlation coefficients (*P* < 0.05). All statistical analyses were performed using STATA software (version 16.0 for Windows; STATA Corp. LLC, Lakeway Drive College Station, TX, USA).

## 3. Results

### 3.1. Stroke Events and Ambient Temperatures in the Studied Areas

In total, 40,315 and 72,073 IS events were identified in Kaohsiung City and the Taipei Area, respectively, from the NHIRD. Table 1 shows that the event rates were 3.09 ± 0.09/1000/year and 2.25 ± 0.04/1000/year in Kaohsiung City and the Taipei Area, respectively. After adjustment for the numbers of permanent residents stratified by age (middle-aged and elderly groups), the IS events were 0.87 ± 0.06/1000/year in the middle-aged group and 12.86 ± 0.51/1000/year in the elderly group in Kaohsiung City, whereas the events were 0.63 ± 0.02/1000/year and 10.18 ± 0.17/1000/year in the same age groups in the Taipei Area. There were statistically significant differences in stroke events between these two areas for each age group. During the study period, 30,023 and 53,090 ICH events occurred in Kaohsiung City and the Taipei Area, respectively. The occurrence rates were 2.30 ± 0.12/1000/year and 1.66 ± 0.08/1000/year in Kaohsiung City and the Taipei Area, respectively. The occurrence rates were 1.25 ± 0.08/1000/year in the middle-aged group and 6.93 ± 0.80/1000/year in the elderly group in Kaohsiung City and 0.84 ± 0.03/1000/year and 5.65 ± 0.41/1000/year in the same age groups in the Taipei Area. There were also statistically significant differences between these two areas for each age group. The daily mean temperatures in Kaohsiung City were significantly higher than those in the Taipei Area, 25.38 ± 3.98 °C vs. 23.26 ± 5.55 °C. The daily temperature differences (the difference between the daily highest and lowest temperatures) were higher in the Taipei Area than those in Kaohsiung City, 5.85 ± 2.67 °C vs. 5.59 ± 1.64 °C. The coefficient variations of daily temperature difference were lower in Kaohsiung City than those in the Taipei Area, 29.0% vs. 45.6%.

### 3.2. Monthly Changes of Stroke Events

Figure 2 shows the monthly stroke events during the six years. The events of IS were higher than those of ICH in the middle-aged and elderly groups in Kaohsiung City (a). Similar findings were also found in the Taipei Area (b). Moreover, there were faint peaks for ICH events in all age groups in both areas and IS events in the elderly group in the Taipei Area in the cold season around months 12, 24, 36, 48, 60, and 72.

### 3.3. Association between Mean Ambient Temperature and Monthly Stroke Events

We used Spearman’s rank correlation coefficients to assess the correlation between the monthly mean temperatures and stroke events. Table 2 shows significant correlations between the temperatures and ICH events in the middle-aged and elderly groups in both studied areas. However, there was only a significant correlation in the elderly IS group in the Taipei Area. Only one correlation coefficient of >0.5 was determined, which suggests a moderate correlation between the monthly mean temperatures and ICH events in the elderly group in the Taipei Area; nevertheless, there were several significant correlations between temperature and stroke events in both areas.

### 3.4. Decompositions of the Daily Mean Temperatures and Stroke Events

We used EEMD to decompose the daily mean temperatures and stroke events. For example, Figure 3 shows the IMFs of the daily mean temperatures (a) and events of ICH in the elderly group (b) in the Taipei Area. There were several obvious periodical signals for IMFs 4–8. To check the consistency of the periodical signals, we defined interpeak periods (IPP) as the intervals between two consecutive peaks. The consistency of IPPs was estimated by coefficient variance (CV) as (SD/mean) × 100%. Table 3 shows the IPPs of IMFs 4–8 of daily ambient temperatures and stroke events in these two studied areas. These data clearly show that the IPPs around 360 days had the lowest CV, except for IMF8 of IS in the middle-aged group and IMF7 for ICH in the elderly group in the Taipei Area. The six-period IMFs were presented in all decomposed data, as shown in Figure 4. After decomposing the original data, we used Spearman’s rank correlation coefficients to examine the correlation between relevant IMFs, which consisted of similar IPPs of ambient temperatures and stroke events. For example, Figure 5 shows the relevant IMFs of ambient temperatures and events of ICH and the correlation coefficients of the IMFs in the elderly group in the Taipei Area. There was a high correlation coefficient between IMF6 of the daily mean temperatures and IMF7 of ICH events.

### 3.5. Temporal Association between the Relevant IMFs of Daily Mean Temperatures and Stroke Events

We used Spearman’s rank correlation coefficients to assess the similarity of relevant IMFs of daily mean temperatures and stroke events in Kaohsiung City and the Taipei Area. Table 4 shows the significant correlations between all IMFs except IMF4 of the daily mean temperatures and IS events in the elderly group and IMF5 of the daily mean temperatures and IS events in the middle-aged group in Kaohsiung City. The correlation coefficients were >0.5 between all IMFs of temperature and ICH in all age groups and for the IS in the elderly group in both areas. Of note, the coefficients were higher in the Taipei Area than in Kaohsiung City.

**Figure 3 jcm-10-05041-f003:**
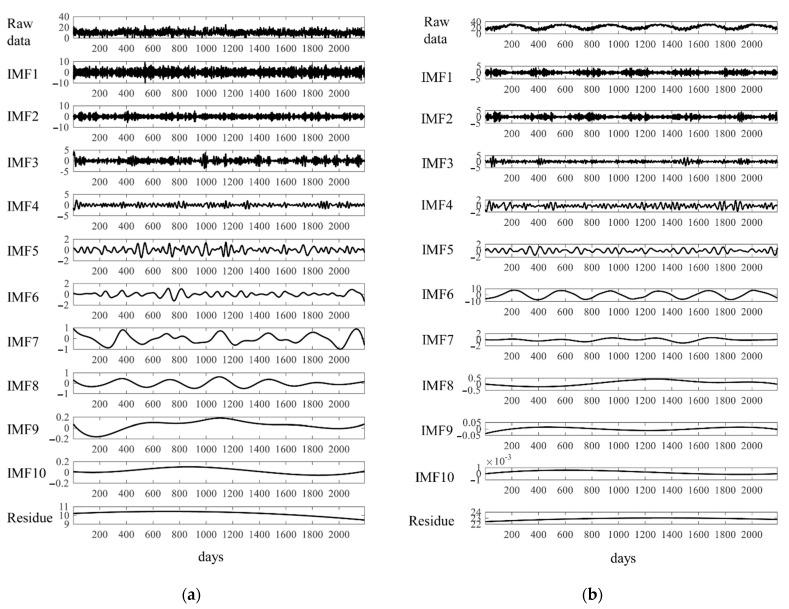
Intrinsic mode functions (IMFs) of decomposed daily mean temperatures (**a**) and events of intracerebral hemorrhage in the elderly group (**b**) in Taipei Area.

**Table 3 jcm-10-05041-t003:** Interpeak periods of decomposed daily ambient temperatures and stroke events in Kaohsiung City and the Taipei Area.

	Kaohsiung					Taipei				
		Ischemic Stroke		Intracerebral Hemorrhage			Ischemic Stroke		Intracerebral Hemorrhage	
	Temp	Middle-Aged	Elderly	Middle-Aged	Elderly	Temp	Middle-Aged	Elderly	Middle-aged	Elderly
IMF4	29.22 ± 8.66	23.84 ± 6.63	23.16 ± 6.14	22.14 ± 6.62	22.39 ± 5.77	28.01 ± 7.69	24.52 ± 6.59	23.67 ± 6.36	22.74 ± 5.25	24.09 ± 7.93
	(29.64)	(27.81)	(26.51)	(29.90)	(25.77)	(27.45)	(26.88)	(26.87)	(23.09)	(32.92)
IMF5	56.92 ± 12.93	49.77 ± 14.96	46.04 ± 11.68	43.47 ± 12.29	46.80 ± 13.32	60.29 ± 13.92	45.94 ± 12.06	42.43 ± 11.83	46.26 ± 14.13	48.30 ± 13.27
	(22.72)	(30.06)	(25.37)	(28.27)	(28.46)	(23.09)	(27.62)	(27.88)	(30.54)	(27.47)
IMF6	195.00 ± 127.45	87.00 ± 34.27	94.95 ± 18.48	89.35 ± 21.57	85.71 ± 27.13	365.80 ± 10.76	97.70 ± 26.13	90.30 ± 24.22	90.73 ± 22.31	95.67 ± 25.76
	(65.36)	(39.39)	(19.46)	(24.14)	(31.65)	(2.94)	(26.75)	(26.82)	(24.59)	(26.93)
IMF7	386.40 ± 32.75	189.20 ± 56.92	171.22 ± 38.98	195.78 ± 47.61	258.57 ± 94.79	361.50 ± 4.37	190.3 ± 33.50	200.80 ± 65.12	219.75 ± 84.36	300.83 ± 81.09
	(8.48)	(30.08)	(22.77)	(24.32)	(36.66)	(1.21)	(17.60)	(32.43)	(38.39)	(26.96)
IMF8	808.00 ± 623.67	357.20 ± 50.34	392.00 ± 19.22	400.75 ± 46.67	383.00 ± 48.05	1425 *	305.83 ± 185.09	375.5 ± 37.74	367.49 ± 5.89	678.00 ± 120.21
	(77.19)	(14.09)	(4.90)	(11.65)	(12.55)		(60.52)	(10.05)	(1.60)	(17.73)

Data of the peak–peak intervals are presented as mean ± SD (days) from IMFs 4 to 8 of the decomposed data. Correlation variance as (SD/mean) × 100% are quoted below the values. Temp: daily mean temperatures. * There are only two peaks with one interpeak interval of IMF8 of the daily mean temperature in Taipei Area.

**Figure 4 jcm-10-05041-f004:**
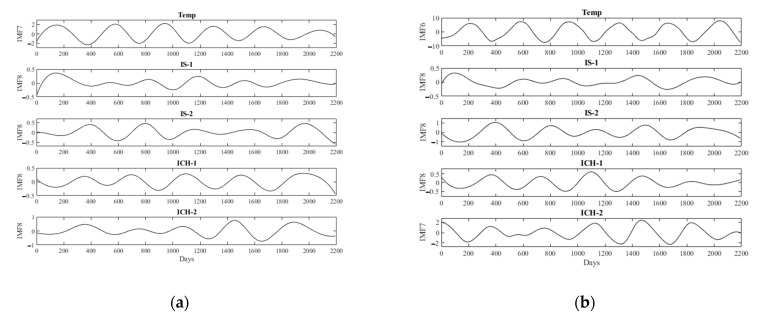
Six-period IMFs of the decomposed data in Kaohsiung City (**a**) and Taipei Area (**b**). Temp_Mean: daily mean temperature, IS-1: ischemic stroke in the middle-aged group, IS-2: ischemic stroke in the elderly group, ICH-1: intracerebral hemorrhage in the middle-aged group, ICH-2: intracerebral hemorrhage in the elderly group.

**Figure 5 jcm-10-05041-f005:**
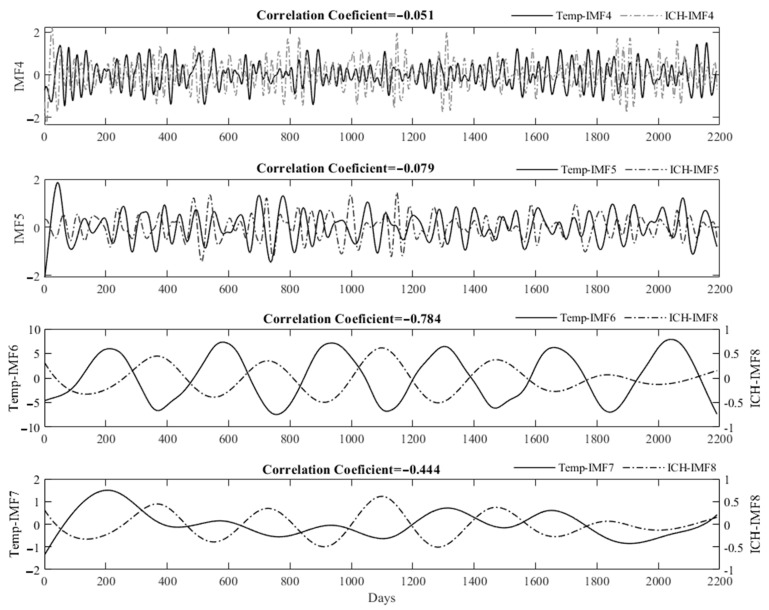
Spearman’s rank correlation coefficients between relevant IMFs of daily events of ICH and mean temperatures and in the elderly group in Taipei Area.

**Table 4 jcm-10-05041-t004:** Spearman’s rank correlation coefficients between the IMFs of temperature and the IMFs of stroke events with similar period intervals in Kaohsiung City and Taipei Area.

		Ischemic Stroke		Intracerebral Hemorrhage	
	Temp	Middle-Aged	Elderly	Middle-Aged	Elderly
Kaohsiung City	IMF4/IMF4	0.098 *	0.024	−0.085 *	0.116 *
	IMF5/IMF5	0.026	0.087 *	−0.080 *	0.070 *
	IMF6/IMF7	−0.246 *	−0.353 *	−0.351 *	**−0.792 ***
	IMF7/IMF8	−0.076 *	**−0.634 ***	**−0.661 ***	**−0.705 ***
Taipei Area	IMF4/IMF4	−0.204 *	−0.137 *	−0.051 *	−0.085 *
	IMF5/IMF5	−0.101 *	−0.149 *	−0.079 *	−0.068 *
	IMF6/IMF8	0.063 *	**−0.585 ***	**−0.784 ***	**−0.883 §***
	IMF7/IMF8	−0.126 *	**−0.552 ***	−0.444 *	**−0.544 §***

IS-1: ischemic stroke in the middle-aged group, IS-2: ischemic stroke in the elderly group, ICH-1: intracerebral hemorrhage in the middle-aged group, ICH-2: intracerebral hemorrhage in the elderly group. §: the correlation with IMF 7 of decomposed stroke incidence. *: *P* < 0.05. Boldface: the correlation coefficient > 0.5.

## 4. Discussion

Table 1 shows typical differences in ambient temperatures between the tropical and subtropical areas, demonstrating higher daily mean temperatures and smaller daily temperature differences and variations in Kaohsiung City compared with the Taipei Area. Although Taiwan has a comprehensive NHIRD and is a unique geographic location, few studies have examined the impact of geographical differences on stroke incidence in Taiwan. Hu et al. reported that the age-adjusted first stroke incidence was 3.21 cases/1000 in Southern Taiwan and 2.95 cases/1000 in Northern Taiwan in a prospective cohort study of 8562 subjects from 1986–1990 [32]. In this study, we used events instead of the incidence or first-ever stroke as an indicator of stroke occurrence; therefore, our data were more relevant than those in previous studies, especially for the events in the elderly group. The interesting coincidental findings about temperature and stroke events found in these two studies may be not merely related to the climates but also attributed to the difference of risk factors of stroke such as hypertension, diabetes, sex, etc.

Although several risk factors including age, sex, hypertension, and diabetes were not analyzed in the current study, our results clearly demonstrate that annual IS and ICH events were higher in Kaohsiung City than in the Taipei Area. Those are similar to previous hospital-based studies among the Chinese. Those showed that IS was more common than ICH among Chinese people. The IS:ICH ratio was 1.59 to 2.31 in community-based studies and much higher in hospital [33]. Our results demonstrated that the IS to ICH ratios were 1.34 and 1.36 in Kaohsiung City and the Taipei Area. After age stratification, the current study found that the ratios were 0.70 and 0.75 in the middle-aged groups and 1.86 and 1.80 in the elderly groups in Kaohsiung City and the Taipei Area, respectively. These results indicate that ICH events are more common than IS events in younger subjects. Similar results were found in a survey of a nationwide insurance database in South Korea [9]. The possible mechanism remains uncertain, and related articles are unavailable.

The association between ambient temperature and stroke incidence remains uncertain. A meta-analysis that recruited 19,736 stroke patients, including 14,199 IS and 3798 ICH, from 26 articles suggested that IS and ICH have a negative correlation with ambient temperature [16]. Although Figure 2 shows a tendency for a reverse relationship between the monthly ambient temperature and stroke events, there was a mild to moderate correlation between ICH events and ambient temperature. Only IS occurrences had a strongly reverse association with ambient temperature in the elderly group in the Taipei Area by Spearman’s rank correlation coefficient (Table 2). The seasonality of the occurrences of IS was not demonstrated in the NHIRD during the period 1998 to 2003 [34].

Our raw data show the daily events of stroke were around 0–30 events/day with irregularity. EEMD is a powerful analytic tool when dealing with >2000 tiny irregular data points after adding adequate white noise. This method can isolate and extract physically meaningful IMFs from the original signals. For example, EEMD demonstrated the El Niño–southern oscillation events by decomposing sea-level atmospheric pressure and temperature (Wu and Huang, 2009) [31]. This method was also used to study the association between environmental or meteorological factors and monthly suicide rates or headaches [35], or to predict COVID-19 endemics [36].

In the current study, for example, the EEMD decomposed the daily mean temperatures and events of ICH of the middle-aged group in the Taipei Area into several IMFs. There were 6-period IMF6 and IMF7 in the decomposed data, respectively (Figure 3). Although EEMD can decomposed intermixed oscillating intermixed signals into several mono-component intrinsic mode functions, the periods of IMFs of temperature and stroke events in all groups are not consistent (Table 3). The meaning of each IMF is not certain, except the IMF1, which is the added white noise. We had found high-frequency rhythm in IMF2 and IMF3, not shown in Table 3 and the text. These high-frequency IMFs may be the daily or weekly periodic changes of the signals. However, they are not consistent and do not have good correlation between each the ambient temperature and stroke events. There are 6-period IMFs at intervals of around 360 days, which were the most consistent decomposed IMFs of all the raw data in both studied areas (Table 3, Figure 4). As an example, these 6-period IMFs of daily events of ICH had good correlation coefficients with the daily mean temperatures in the Taipei Area (Figure 5). Table 4 summarizes the correlation coefficients of decomposed IMFs of daily stroke events with the IMF of mean temperatures with similar intervals in Kaohsiung City and the Taipei Area. There were reverse associations between daily stroke events and mean temperatures. These were more prominent between the daily events of ICH in any age group and the daily occurrences of IS in the elderly group in both studied areas. Similar results were found in a meta-analysis, which revealed that elderly subjects had an increased incidence of IS and ICH in cold weather [15].

There were statistically significant correlations between temperature and stroke events in most groups. Moderate-to-strong correlations, as coefficient > 0.5 [37], with temperature were found in the 5–6-period IMFs of decomposed ICH in both age groups and IS in the elderly group in Kaohsiung City and the Taipei Area by Spearman’s rank correlation. Of note, the correlations were stronger in the Taipei Area (subtropical area) than in Kaohsiung City (tropical area). A similar method was applied to study the spatial–temporal association of environmental factors, including temperature, soil moisture, and photosynthetically active radiation [38].

Our data show obvious correlations between ambient temperature and stroke events, which were more prominent in the subtropical area. These results differ from those in the meta-analysis study, which suggested that the associations decreased with latitude [16]. Unlike the meta-analysis, the current study used a cohort in two areas at different latitudes and climate zones from a big dataset of same healthcare system. Our results were more relevant to the situation in the real world. Nevertheless, both studies demonstrated that associations between ambient temperature and stroke incidence were more apparent in ICH.

A limitation of the current study was that the risk factors of stroke, which were not included in our initial data mining, were not analyzed. We propose that remote risk factors may not influence the dynamic daily changes of stroke events. Therefore, our study showed a clear temporal association between stroke events and ambient temperature in tropical and subtropical areas. Meanwhile, we could not analyze the differences of risk factors between these two studied areas. Those may be confounding or major factors influencing the differences of the associations between Kaohsiung City and Taipei Areas. The data of risk factors were not available in our original databank, and relevant articles are not available in the literature either. Nevertheless, based on the nationwide equal healthcare systems and similar age distribution between these two studied areas, our study still offered a naval and useful information about the possible influences of altitude on the association between ambient temperature and stroke events as compared with the current meta-analyses.

## 5. Conclusions

Our study of the nationwide insurance claim data shows that the events of stroke have a tendency to be reversely associated with ambient temperature. The correlations became obvious after the nonlinear decomposition by EEMD. These may suggest that cold weather is a risk factor of stroke events. The method to demonstrate these temporal associations can hopefully promote further studies about similar dynamic associations between environmental factors and occurrences of stroke, myocardial infarction, endemic and pandemic diseases, etc. Meanwhile, the correlation coefficients were higher in the subtropical area than those in the tropical area. The importance of 360 day IMFs of stroke events and possible influence of latitude or climate on daily stroke events has not been reported in the past. These offer potential research prospects on the temporal and geographic associations between environmental factors and disease events by using the unique NHIRD.

## Figures and Tables

**Figure 1 jcm-10-05041-f001:**
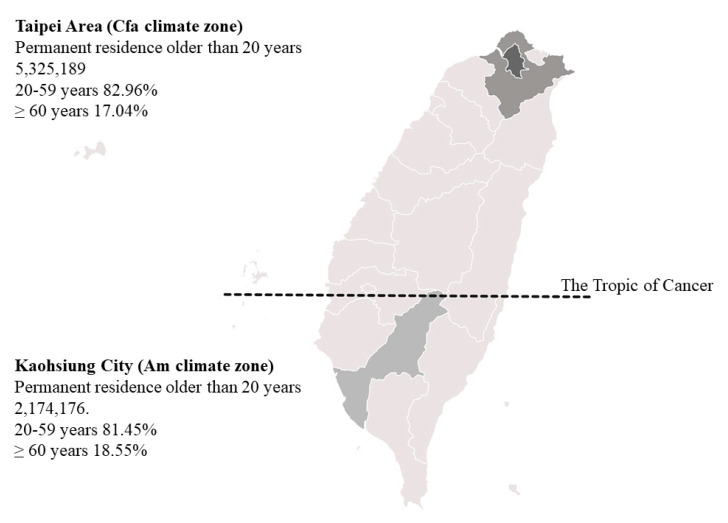
Populations and geographic and climate features of Taiwan. The Tropic of Cancer (23°26′12.6″ north of the Equator) passes through Taiwan and divides this island into tropical (south) and subtropical (north) areas. Taipei and New Taipei locate in a Cfa climate zone, while Kaohsiung City is situated in an Am climate zone.

**Figure 2 jcm-10-05041-f002:**
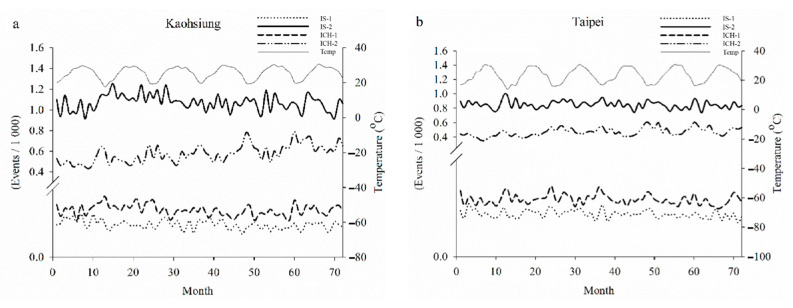
(**a**) Monthly events of ischemic stroke in the middle-aged group (IS-1, black dotted line) and the elderly group (IS-2, black line); intracerebral hemorrhage in the middle-aged group (ICH-1, black dashed line) and the elderly group (ICH-2, black dashed and dotted line) and the temperature (gray line) in Kaohsiung City. (**b**) Monthly events of ischemic stroke in the middle-aged group (IS-1, black dotted line) and the elderly (IS-2, black line); intracerebral hemorrhage events in the middle-aged group (ICH-1, black dashed line) and the elderly group (ICH-2, black dashed and dotted line) and temperature (gray line) in Taipei Area.

**Table 1 jcm-10-05041-t001:** Daily mean temperatures, temperature differences, and stroke events in the Kaohsiung City and Taipei Area.

	Kaohsiung	Taipei
Daily mean temperatures (°C)	25.38 ± 3.98	23.26 ± 5.55 *
Daily temperature differences (°C)	5.59 ± 1.62	5.85 ± 2.67 *
Ischemic stroke(events/1000/year)	3.09 ± 0.09	2.25 ± 0.04 *
Middle-aged	0.87 ± 0.06	0.63 ± 0.02 *
Elderly	12.86 ± 0.51	10.18 ± 0.17 *
Intracerebral hemorrhage (events/1000/year)	2.30 ± 0.12	1.66 ± 0.08 *
Middle-aged	1.25 ± 0.08	0.84 ± 0.03 *
Elderly	6.93 ± 0.80	5.65 ± 0.41 *

Data are presented as mean ± SD. Significance of differences determined by the nonparametric Mann–Whitney U-test as * *P* < 0.05.

**Table 2 jcm-10-05041-t002:** Spearman’s rank correlation coefficients between monthly mean temperatures and stroke events.

Stoke Events	Kaohsiung	Taipei
Ischemic Stroke		
Middle-aged	−0.037	0.019
Elderly	−0.185	−0.485 *
Intracerebral hemorrhage		
Middle-aged	−0.294 *	−0.464 *
Elderly	−0.367 *	**−0.571 ***

*: *P* < 0.05. Boldface: the correlation coefficient > 0.5 considered to have moderate association.

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
