# Peer review of "Dynamic Changes and Temporal Association with Ambient Temperatures: Nonlinear Analyses of Stroke Events from a National Health Insurance Database"

_jcm, 2021, doi:10.3390/jcm10215041_

Round 1
Reviewer 1 Report
The authros reported an interesting study about the association between stroke incidence and temparature by dividing areas into the subtropical areas and the tropical area in Taiwan.
As they mentioned, the association between ambient temperature and stroke incidence remains controversial.
1. I agree that this study has suitable model to evaluate the relationship between ambient temperature and stroke incidence using National Health Insurance Research Database.
2. It would be better for the letters in Table 2 be clearer.
3. Why did not this study include other risk factors of stroke, such as age, sex, hypertension, and diabetes, which could be available in NIHRD? It woud be better to discuss this, although they had already mentioned it in limitation briefly.
4. In a study comparing two different cities like this study, it is necessary to compare the baseline characteristics between two cities. For example, is there any difference in age or sex between peopel in the two cities?
5. Additional English correction is not required.
Author Response
The reply was summarized as the attachment.

Reviewer 2 Report
Summary
The manuscript provides a creative and unique look at the relationship between temperature and stroke incidence using the separation of 2 urban areas in Taiwan between subtropical and tropical climate zones. They find higher overall stroke rates in the tropical region than in the subtropical region. Using a nonlinear analysis method they also find a significant negative correlation between mean daily temperatures and seasonal variations in stroke rates.
Comments
- The manuscript provides a broad background review of published information about associations of temperature with stroke rates. The study provides a careful methodological approach and a novel quantitative analytic approach, with ensemble empirical mode decomposition (EEMD).
- EEMD is likely unfamiliar to most of the readership of this clinical journal, and the authors should provide more help in understanding the method and especially in interpretation of its results. For instance, is there specificity of the numbered Intrinsic mode functions in terms of pre-determined temporal frequencies? By scan of Figure 3, it seems clear that the dominant frequency of the IMF decreases in the higher-numbered functions, but this is not explained.
- Similarly, in the discussion, the authors should provide more help in providing possible interpretation of the significance of the numerical results. For instance, do the results answer questions such as whether there is a more important role for short-term changes in temperature on a daily or weekly basis, or for of seasonal variations on a monthly basis?
- The overall results are notable in suggesting an apparent contrast between the direction of association between average temperature and stroke rates (positive) and that of daily or seasonal variation in temperature and stroke rates (negative, with higher stroke rates during the lower temperatures of winter season). This interesting anomaly or contrast is worthy of discussion but is not addressed. The authors should offer some reflections and interpretation of this.
- On page 4, line 175, the average daily temperature differences are listed in reverse order, when checked against Table 1.
- The font in the legend of Figure 2 is too small.
Author Response

(The authors gave the same response as above.)
